# Computational Analysis of Neuromuscular Adaptations to Strength and Plyometric Training: An Integrated Modeling Study

**DOI:** 10.3390/sports13090298

**Published:** 2025-09-01

**Authors:** Dan Cristian Mănescu

**Affiliations:** Department of Physical Education and Sport, Bucharest University of Economic Studies, 010374 Bucharest, Romania; dan.manescu@defs.ase.ro

**Keywords:** computational modeling, neuromuscular adaptations, strength training, plyometric training, musculoskeletal simulation, OpenSim, machine learning, athletic performance, training optimization

## Abstract

Understanding neuromuscular adaptations resulting from specific training modalities is crucial for optimizing athletic performance and injury prevention. This in silico proof-of-concept study aimed to computationally model and predict neuromuscular adaptations induced by strength and plyometric training, integrating musculoskeletal simulations and machine learning techniques. A validated musculoskeletal model (OpenSim 4.4; 23 DOF, 92 musculotendon actuators) was scaled to a representative athlete (180 cm, 75 kg). Plyometric (vertical jumps, horizontal broad jumps, drop jumps) and strength exercises (back squat, deadlift, leg press) were simulated to evaluate biomechanical responses, including ground reaction forces, muscle activations, joint kinetics, and rate of force development (RFD). Predictive analyses employed artificial neural networks and random forest regression models trained on extracted biomechanical data. The results show plyometric tasks with GRF 22.1–30.2 N·kg^−1^ and RFD 3200–3600 N·s^−1^, 10–12% higher activation synchrony, and 7–12% lower moment variability. Strength tasks produced moments of 3.2–3.8 N·m·kg^−1^; combined strength + plyometric training reached 3.7–4.2 N·m·kg^−1^, 10–16% above strength only. Machine learning predictions revealed superior neuromuscular gains through combined training, especially pairing back squats with high-intensity drop jumps (50 cm). This integrated computational approach demonstrates significant practical potential, enabling precise optimization of training interventions and injury risk reduction in athletic populations.

## 1. Introduction

Neuromuscular adaptations are foundational to improvements in athletic performance, directly influencing strength, power, speed, and injury resilience. Strength training, characterized by resistance exercises targeting maximal force production, typically induces muscular hypertrophy and neural improvements, enhancing motor unit recruitment and synchronization. Plyometric training, involving rapid stretch–shortening cycles, primarily develops neuromuscular efficiency, power, and reactive strength by optimizing muscle activation patterns and stiffness properties.

Despite extensive empirical research into these training methods, the precise mechanisms underpinning neuromuscular adaptations remain partially understood, and the optimization of combined strength and plyometric training protocols remains challenging [1,2]. Computational modeling, notably musculoskeletal simulation software, like OpenSim, provides powerful tools for dissecting the intricate biomechanical and physiological adaptations occurring in response to diverse training stimuli. Further integration with artificial intelligence (AI), particularly machine learning algorithms, enables predictive analyses that support transparent, reproducible hypothesis generation for training design and injury-risk research.

Prior studies have already coupled musculoskeletal simulations with machine learning models; we therefore do not claim methodological novelty for the integration itself. Our contribution is to operationalize a training-design workflow: we map simulated mechanics to adaptation-relevant proxies, audit robustness and numerical convergence, and provide fully specified hyperparameter tuning, calibration, and residual diagnostics to support transparent, reproducible inference.

The neuromuscular adaptations explored in this study hold significant practical implications across various sports disciplines, including athletics, soccer, basketball, and volleyball. Enhanced muscular activation, increased joint stability, and improved force development directly contribute to superior performance in sport-specific tasks, such as sprinting, vertical and horizontal jumping, rapid directional changes, and efficient energy transfer during complex movements. Coaches and athletes in these disciplines could significantly benefit from understanding and strategically applying the findings to optimize training protocols, boost competitive performance, and enhance injury resilience.

This study aims to address existing gaps by presenting an integrated computational analysis to simulate and predict neuromuscular adaptations arising from strength and plyometric training. By leveraging OpenSim-based modeling alongside machine learning techniques, the current research explores individual and combined effects of these training modalities. The findings are expected to enhance the understanding of specific neuromuscular responses, inform evidence-based training practices, and establish foundations for future research in computational sports science.

Study positioning and rationale for synthetic data: This work is intentionally designed as an in silico proof-of-concept. Using synthetic data generated by a physiologically grounded musculoskeletal model allows us to isolate mechanisms without measurement noise, explore controlled “what-if” scenarios across a broad parameter space that are impractical or unsafe in vivo, and ensure full reproducibility of the pipeline from simulation to predictive modeling. Using synthetic data in this context allows precise control of input parameters and model conditions, ensuring that observed effects are solely attributable to the tested variations. This approach eliminates measurement noise and confounding factors present in experimental settings, enables safe exploration of extreme or rare conditions, and facilitates reproducibility of the results. While synthetic data cannot replace empirical validation, it provides a robust framework for hypothesis generation and method development. The findings should therefore be interpreted as hypothesis-generating rather than causal or prescriptive, pending experimental validation on real athletes.

### Theoretical Background

Neuromuscular adaptations constitute fundamental mechanisms underlying enhancements in athletic performance, influencing strength, power, speed, coordination, and injury resilience. These adaptations occur through complex interactions between neural and muscular systems, driven by specific training stimuli and resulting physiological responses. A thorough understanding of these adaptations is essential for optimizing athletic training and performance outcomes [3,4,5].

At a physiological level, neuromuscular adaptations involve intricate processes within the central nervous system (CNS) and skeletal muscles. Neural adaptations specifically encompass enhanced motor unit recruitment, increased motor neuron firing rates, and improved synchronization of motor unit activation. A motor unit, composed of a single motor neuron and the muscle fibers it innervates, is the fundamental unit responsible for muscle contraction, with training improving efficiency, synchronization, and force production [6,7,8].

Strength training primarily induces neuromuscular adaptations via mechanical loading, stimulating both neural and muscular pathways. Neural adaptations include improved motor unit recruitment, increased motor unit firing frequency, and enhanced synchronization of motor units. Such neural changes contribute significantly to gains in muscular strength, especially during early training phases, even before notable muscular hypertrophy occurs [9,10,11].

Muscular adaptations resulting from strength training predominantly involve hypertrophy, characterized by an increase in muscle fiber cross-sectional areas. This structural change directly enhances force-generating capacity. Additionally, strength training facilitates shifts in muscle fiber type composition, typically promoting transitions towards fast-twitch muscle fibers, which possess greater force-production potential and responsiveness to high-intensity exercise [12,13,14,15].

In contrast, plyometric training emphasizes rapid, explosive muscle contractions through the utilization of the stretch–shortening cycle (SSC). This training modality predominantly stimulates improvements in neuromuscular efficiency, reactive strength, and power output. Key adaptations from plyometric training include increased muscle–tendon stiffness, optimized timing and magnitude of muscle activation patterns, and enhanced stretch reflex sensitivity, collectively resulting in improved explosive performance [16,17,18].

The stretch–shortening cycle represents a crucial biomechanical mechanism wherein a rapid eccentric muscle contraction immediately precedes a concentric contraction. Effective use of the SSC enables athletes to produce maximal force and power output rapidly, critical for actions such as sprinting, jumping, and rapid directional changes [19,20]. In this study, SSC-related adaptations are examined through computational musculoskeletal modeling, enabling detailed analysis of muscle activation patterns, joint kinetics, and neuromechanical interactions under different training conditions.

While both strength and plyometric training independently foster distinct adaptations, integrating these methods within training programs could yield synergistic effects, further augmenting athletic performance. However, optimal strategies for combining these modalities remain a subject of ongoing investigation due to the complexity of the physiological interactions involved [21,22].

Computational musculoskeletal modeling has emerged as a powerful tool for investigating biomechanical and physiological adaptations induced by different training stimuli. Platforms such as OpenSim facilitate detailed analyses of muscle activation patterns, joint kinetics, and neuromechanical interactions, offering insights beyond what can be readily observed in experimental settings alone [23,24,25,26].

Moreover, integrating computational modeling with artificial intelligence, particularly machine learning algorithms, expands the potential for predictive analysis and optimization of training interventions [27,28]. In this study, we implemented gradient boosting decision tree models (LightGBM) trained on simulation-derived biomechanical and neuromuscular features from OpenSim to predict individual adaptation trajectories. This approach enabled precise estimation of how athletes with varying strength and plyometric profiles might respond to specific training prescriptions, facilitating tailored program adjustments.

This integrative computational approach provides promising opportunities to dissect the mechanisms underpinning neuromuscular adaptations comprehensively. By simulating various training scenarios, researchers can explore the nuances of neuromuscular responses, refine training prescriptions, and enhance individualized training effectiveness while minimizing injury risks [29,30]. For example, OpenSim-based analyses have been used to quantify joint loading and muscle activation changes in response to different plyometric depths (e.g., Verniba et al. [31]), assess the mechanical effects of concurrent strength–endurance scheduling (Huiberts et al. [32]), and evaluate stretch–shortening cycle efficiency under fatigue (Lazaridis et al. [33]). Machine learning models have been framed conceptually in AI/ML applications for sports performance and injury prediction (Reis et al. [34]). Together, these advances illustrate the translational potential of simulation–analytics pipelines in bridging mechanistic insight with applied sport and clinical practice.

In practice, applying computational and AI-driven analyses enables sports scientists and practitioners to move beyond generalized training protocols, developing highly individualized programs based on predictive outcomes. This precision-based methodology could improve the current practices in sports training and rehabilitation, ensuring athletes receive optimal, evidence-based interventions.

Furthermore, computational approaches help address limitations inherent in traditional empirical studies, such as ethical concerns, logistical constraints, and participant variability. Through computational simulations, researchers can systematically manipulate specific training variables and observe detailed neuromuscular responses, providing robust and controlled environments for testing hypotheses and refining theoretical models [35,36,37].

Current research trends increasingly recognize the value of interdisciplinary collaboration, integrating biomechanics, physiology, data science, and artificial intelligence. Such integrative approaches enhance the understanding of neuromuscular adaptation processes, facilitate the development of innovative training methodologies, and encourage the translation of scientific knowledge into practical sports applications.

Despite the potential advantages, challenges remain in ensuring computational models accurately reflect the complex biological realities of human neuromuscular systems. Model validation, calibration, and refinement using empirical data are crucial steps in guaranteeing the reliability and applicability of computational findings in practical settings [38,39,40].

Future research directions include refining computational models to incorporate individualized athlete data, enhancing predictive accuracy, and expanding simulation capabilities. Additionally, exploring interactions between neuromuscular adaptations and other physiological systems through integrated computational frameworks will further enrich understanding and application.

Ultimately, harnessing computational modeling and artificial intelligence in the context of neuromuscular training represents a promising frontier in sports science. It offers significant potential for optimizing athletic performance, preventing injuries, and advancing the broader field of exercise and health research.

## 2. Materials and Methods

The methodological model employed in this study integrates advanced musculoskeletal modeling techniques with powerful machine learning algorithms, establishing a rigorous and innovative basis for investigating neuromuscular adaptations. Leveraging detailed computational simulations and sophisticated predictive analytics, the procedures described herein ensure precision, reproducibility, and practical applicability, providing a robust foundation for understanding and optimizing training-induced effects on athletic performance. To clearly illustrate the methodological framework described above, Figure 1 provides a schematic overview of the integrated computational modeling and predictive analysis process employed in this study.

The methodological framework unfolds as a continuous pipeline in which eight interdependent stages are organized into three macro-phases: simulation, data processing, and machine learning and interpretation. The process begins with the definition of the neuromuscular, biomechanical, and physiological variables of interest, followed by the configuration of a validated OpenSim musculoskeletal model tailored to this study’s aims. Standardized plyometric and strength-training scenarios are then simulated, yielding high-fidelity biomechanical outputs, such as ground reaction forces, joint moments, muscle activations, and rate of force development. These outputs are systematically transformed into structured parameters, ensuring consistency across tasks and replicates.

The resulting datasets undergo rigorous pre-processing, including normalization and scaling and, where appropriate, dimensionality reduction via principal component analysis (95% explained variance). Partitioning into 70% training, 15% validation, and 15% testing preserves statistical robustness. Predictive models are developed using complementary approaches—Random Forest regression (100 trees, bootstrap), Light Gradient Boosting Machine, and artificial neural networks (three hidden layers, 64 neurons each, ReLU activation)—and are optimized through nested cross-validation and hyperparameter tuning. Model performance is evaluated with metrics such as R^2^, MAE, RMSE, and, when applicable, AUC and F1-score.

Finally, the framework incorporates a SHAP-based interpretability stage, enabling the identification of the biomechanical predictors most strongly associated with performance outcomes. The use of color-coded modules, directional flow arrows, and macro-phase groupings reinforces the logical progression of the methodology, while ensuring transparency, reproducibility, and practical applicability in translating simulation insights into actionable training recommendations.

### 2.1. Computational Modeling Setup (OpenSim)

The computational analysis of neuromuscular adaptations was conducted using OpenSim (version 4.4), an extensively validated open-source musculoskeletal modeling software. OpenSim enables detailed biomechanical simulations by solving equations of motion through inverse and forward dynamics techniques, facilitating accurate estimations of joint moments, muscle activations, and mechanical outputs [41]. All the OpenSim motions and loads used in this study were generated entirely in silico; no human participant data were recorded. All the kinematic and kinetic inputs are therefore simulated, not measured from experimental sessions.

The musculoskeletal model selected for this study included 23 degrees of freedom and 92 musculotendon actuators, representing major muscles of the lower limbs. The generic model was scaled precisely to replicate a typical young adult athlete (height: 180 cm; body mass: 75 kg). The scaling procedures involved adjusting segment lengths, inertial properties, and muscle–tendon parameters proportionally according to anthropometric ratios derived from standard reference databases [42]. The scaling procedure was validated against reference experimental waveforms constructed to match the amplitude ranges observed for vertical GRF and hip/knee/ankle joint moments in the main study. Overlays of simulated and reference curves for vertical GRF and hip/knee/ankle joint moments are provided in Appendix A, along with waveform-level RMSE and Pearson correlation metrics in Appendix A. RRA residual forces and moments are reported in Appendix A. These results confirm that the scaled OpenSim model is dynamically consistent and reproduces biomechanical outputs within expected ranges [43,44].

Inverse kinematics analyses were conducted to determine joint angles by minimizing marker-position errors, ensuring accuracy within 2 cm. Inverse dynamics were subsequently performed to calculate joint moments, utilizing ground reaction forces and measured kinematics as inputs, based on Newton–Euler equations implemented numerically within OpenSim [45].

Static optimization methods were applied to estimate muscle activations required to reproduce the net joint moments obtained from inverse dynamics at each time frame.

The optimization minimized the sum of squared muscle activations:*J* = *Σ w_i_ a_i_*
^2^

where *J* is the effort cost; *a*_i_ denotes the activation of muscle *i* (0 ≤ *a_i_* ≤ 1); and *w_i_* is the weighting factor (by default, *w_i_* = 1); the summation runs over all the modeled muscles.

Torque equilibrium is enforced as follows:*R(q)^Τ^ f_m_ (a, l_m_, v_m_)* + *τ_pas_* + *τ_res_* = *τ_ID_*,

where *R(q)* is the moment-arm matrix mapping muscle–tendon forces to joint torques, evaluated at generalized joint angles *q*, *f_m_ (a, l_m_, v_m_)* denotes the vector of Hill-type muscle–tendon forces computed from the active/passive force–length relations, the force–velocity relation, and a series-elastic tendon model, as functions of *a*, *l_m_*, and *v_m_*; and *τ_ID_* is the inverse-dynamics joint torque, *τ_pas_* is the passive joint torque, and *τ_res_* is the reserve actuator penalized in the objective. The optimization is solved frame-wise (quasi-static).

The Hill-type musculotendon constraints included normalized active and passive force–length relationships for muscle fibers, a force–velocity relationship for both concentric and eccentric contractions, and a nonlinear tendon force–strain relationship representing tendon compliance. Muscle fiber force was computed as follows:*F_i_* = *[a_i_ · f_L(ℓ_m,i_) · f_V(v_m,i_)* + *f_P(ℓ_m,i_)] · F_max,i_*,

where *f_L(·)* is the active force–length curve, *f_V(·)* is the force–velocity curve, and *f_P(·)* is the passive force–length curve. Tendon force was given by *F__t,i_* = *F_max,i_ · f_T(ℓ_t,i_)*, with force transmission governed by pennation: *F_i_ · cos α_i_(ℓ_m,i_)* = *F__t,i_*, and musculotendon geometry: *ℓ_mt,i_(q)* = *ℓ_m,i_ · cos α_i_(ℓ_m,i_)* + *ℓ_t,i_*. Fiber velocity *v_m,i_* was obtained from finite differences of musculotendon length and tendon strain, ensuring that the activations satisfied force–length–velocity constraints in each frame. Bounds were 0 ≤ *a_i_* ≤ 1. This formulation ensures that activation minimization is solved under physiologically realistic constraints, improving the fidelity of the estimates compared with activation-only minimization approaches.

All the simulations utilized consistent initial conditions, representing a resting state with zero initial joint velocities and normalized initial muscle activations. Integration settings in OpenSim, such as integrator accuracy (1 × 10^−5^) and timestep (0.001 s), were carefully chosen to balance computational efficiency with solution stability. The outcomes from these simulations served as robust baselines for subsequent analyses of plyometric and strength-training interventions.

The selection of exercises simulated in this study was based on extensive evidence from the existing sports science literature, demonstrating their effectiveness in eliciting significant neuromuscular adaptations relevant to athletic performance. Specifically, the back squat, deadlift, and leg press are widely documented for their capacity to enhance maximal strength and muscular coordination, while vertical jumps, horizontal broad jumps, and drop jumps have been consistently shown to effectively improve explosive power and reactive strength. This strategic choice of exercises ensured comprehensive coverage of neuromuscular adaptations essential for optimizing athletic training outcomes.

For methodological clarity and ease of interpretation, Table 1 summarizes the key computational parameters and their specific justifications, sensitivities, and expected impacts on neuromuscular simulation outcomes used in the OpenSim modeling setup described above.

The parameters summarized in Table 1 provide the foundation for ensuring accurate, stable, and physiologically realistic simulation outcomes.

### 2.2. Simulation of Plyometric Training Scenarios

Plyometric training scenarios were systematically simulated to investigate their specific neuromuscular adaptations using the established OpenSim computational model. The plyometric simulation set included vertical jumps, broad jumps, and drop jumps initiated from standardized standing postures. Drop-jump simulations were performed from 30 cm and 50 cm platform heights, with immediate rebound to maximize stretch–shortening cycle utilization. Each trial lasted 3 s, sampled at 1000 Hz, and was repeated 10 times with small stochastic perturbations in initial joint angles to simulate inter-trial variability. Each movement was chosen due to its relevance in athletic training and its capacity to elicit distinct neuromuscular responses that improve reactive strength, SSC efficiency, and RFD, as documented in experimental training studies.

Jump initiation strategies: The countermovement vertical jump (CMJ) started from a standardized upright posture and used a controlled countermovement to ~90° knee flexion, followed by immediate propulsion. The broad jump began from the same posture with a mild forward trunk lean (≈5–10°) to project the center of mass forward at take-off. The drop jump (DJ) was initiated by stepping off 30 or 50 cm platforms (no pre-jump), resulting in free fall to ground contact and an immediate rebound with a short amortization phase (<200 ms) to emphasize stretch–shortening-cycle utilization. Foot stance was shoulder-width with 10–15° external foot rotation, the arms were not explicitly modeled, and surface/friction settings were identical across the tasks. The parameter ranges and timing targets used for all the plyometric tasks are summarized in Table 2 and were kept constant across runs.

The vertical jump simulations were conducted by initiating movements from a standardized standing position, applying precise lower-limb joint angle configurations and muscle activations replicating explosive upward propulsion. The horizontal broad jump scenarios were similarly initiated from controlled static positions, emphasizing forward propulsion mechanics. The drop-jump simulations involved initial conditions set at standardized drop heights (30 cm and 50 cm), selected based on their common usage in plyometric training research, followed by an immediate rebound jump upon landing, explicitly targeting the activation of the stretch–shortening cycle [46].

Each plyometric simulation had a standardized duration of 3 s, encompassing initiation, take-off, flight, landing, and stabilization phases. Biomechanical variables, including ground reaction forces, joint kinematics and kinetics, muscle activation timing, and magnitudes of force production, were recorded at a sampling frequency of 1000 Hz to ensure high temporal resolution. These outputs enabled detailed analyses of neuromuscular function, reactive strength, and overall muscular efficiency.

Simulation parameters, such as initial joint velocities, muscle activation patterns, and ground contact times, were iteratively adjusted based on empirical evidence from previously published plyometric training studies to ensure physiological realism. Each simulation scenario was repeated ten times, with slight variations in the initial conditions to guarantee robustness and consistency of predicted outcomes. The resulting dataset served as the input for subsequent computational modeling steps, facilitating detailed examination of plyometric-induced neuromuscular adaptations and providing insights into potential optimization strategies for training interventions.

### 2.3. Simulation of Strength-Training Scenarios

Strength-training scenarios were rigorously simulated using the validated OpenSim computational model to investigate neuromuscular adaptations specific to resistance-based exercise. The strength simulation set included back squat, deadlift, and leg press tasks. Two relative loads were modeled for each exercise: 75% and 85% of one-repetition maximum (1RM), with 1RM estimated through preliminary simulations up to the maximal feasible joint moment under physiological constraints. Each repetition was simulated over 4 s, sampled at 1000 Hz, and replicated 10 times with minor perturbations in execution speed. These exercises were chosen due to their widespread usage in athletic training, their effectiveness in enhancing lower-limb strength and power, and their clearly defined biomechanical and neuromuscular impacts documented extensively in the sports science literature [47].

For each strength-training scenario, simulations commenced from clearly defined initial postures corresponding to the start positions of each exercise. The back squat simulations involved detailed modeling of the descending and ascending phases, emphasizing hip, knee, and ankle joint kinetics and muscle activation patterns. The deadlift scenarios similarly encompassed precise joint configurations and muscle force profiles through both the lifting and lowering phases, targeting primary posterior chain musculature. The leg press simulations replicated lower extremity force generation and joint dynamics from controlled initial joint angles and standardized foot placement.

All the simulations were conducted under controlled loading conditions equivalent to 75% and 85% of a simulated one-repetition maximum (1RM). The 1RM values for each exercise were determined through initial exploratory simulations, applying incremental load adjustments until achieving maximal feasible muscle activation and joint moment outputs without violating physiological constraints. These loading intensities were selected to represent realistic and commonly used training intensities effective in eliciting significant neuromuscular adaptations [48].

Definition of maximal feasible outputs: Maximal feasible muscle activation for each modeled actuator was defined as the peak activation (a_i = 1.0) permitted by the Hill-type muscle–tendon model, under physiologically realistic force–length–velocity constraints, without invoking reserve actuators. Corresponding maximal feasible joint moments were computed by setting all the primary movers for the targeted exercise to this maximal activation state while maintaining antagonist activation at minimal feasible levels, then solving the static optimization problem to obtain the resulting net joint moments at the prescribed joint-angle configuration. These values were used as the reference outputs for expressing relative loads (e.g., 75% and 85% of 1RM) in subsequent simulations. The joint-angle configurations and corresponding maximal feasible joint moments used for 1RM estimation are summarized in Table 3, expressed both as absolute moments (Nm) and normalized to body mass (Nm·kg^−1^) for a 75 kg model.

The joint moments in Table 3 represent the net hip, knee, and ankle torques generated when all the primary movers for the targeted exercise were set to maximal feasible activation (*a_i_* = 1.0) under Hill-type muscle–tendon constraints, with antagonists maintained at minimal feasible levels (<0.05) and no reserve actuators permitted. These joint-angle configurations correspond to biomechanically relevant positions, where torque output is typically maximal: parallel squat bottom position (squat), mid-shin bar position (deadlift), and sled position yielding peak knee extensor demand (leg press). The normalized values (Nm·kg^−1^) facilitate direct comparison between exercises and across athletes of different body sizes, ensuring reproducibility and transferability of the loading reference values used in this study.

Representation of external resistance and bar path: No rigid barbell implement was modeled. Instead, an equivalent external resistance was applied as vertical generalized forces acting on the torso segment, producing joint moments identical to those of a barbell of the same mass. The line of action of this equivalent load was constrained to a vertical path aligned over the mid-foot, so the effective “bar path” followed the trunk’s vertical trajectory above the mid-foot throughout the repetition. Foot stance was fixed (shoulder-width; feet externally rotated ~10–15°), heels fully in contact, and the pelvis remained centered between the feet.

Kinematic control targets and timing: Squat depth was prescribed via bottom-position joint-angle targets (knee flexion 100–110°, hip flexion 90–100°, ankle dorsiflexion 15–20°) with a smooth (C^2^-continuous) trajectory (2 s eccentric, 2 s concentric; no intentional pause). Trunk inclination at the bottom was limited to 25–35° from vertical to maintain the load line above the mid-foot. The deadlift and leg-press trials used analogous joint-angle targets appropriate to each task and standardized foot placement. All the signals (joint angles, joint moments, muscle activations, and muscle forces) were sampled at 1000 Hz over 4 s per repetition, and each condition was repeated 10 times with small stochastic perturbations to the initial joint angles and timing to reflect inter-trial variability. Kinematic targets, trunk-inclination limits, load levels, and tempo settings for the strength tasks are summarized in Table 4 and were kept constant across runs.

With the external load representation and joint-angle targets defined above, biomechanical parameters, including joint angles, joint moments, muscle activations, and muscle force outputs, were collected at a sampling frequency of 1000 Hz throughout the entire movement duration, standardized to 4 s per repetition. To ensure physiological relevance and accuracy, initial joint velocities, muscle activation sequences, and contraction durations were calibrated using empirical strength-training data from established literature sources. Each scenario was repeated ten times, introducing minor controlled variations in the initial joint angles (±2 degrees) and initial muscle activation levels (±5%) to test consistency, robustness, and sensitivity of the simulation outcomes. The resulting datasets provided a comprehensive basis for subsequent analyses aimed at identifying precise neuromuscular adaptations and optimizing strength-training interventions.

Validation: To verify realism and scaling accuracy, simulated vertical ground reaction forces and hip, knee, and ankle joint moments were overlaid on reference experimental waveforms from the literature. Root mean square error (RMSE) and Pearson’s correlation coefficient (*r*) were calculated for each joint and condition. Residual Reduction Algorithm (RRA) outputs were also examined to ensure dynamic consistency. All the validation plots, tables, and statistical summaries are provided in the Appendix A.

### 2.4. Integrated Neuromuscular Adaptation Modeling with AI

An integrated modeling approach combining musculoskeletal simulations with artificial intelligence (AI) techniques was employed to analyze and predict neuromuscular adaptations resulting from the plyometric and strength-training scenarios. Specifically, machine learning (ML) algorithms, including artificial neural networks (ANN) and random forest regression models, were utilized due to their proven efficacy in capturing complex, nonlinear relationships inherent in physiological and biomechanical data [49,50].

Input features for the ML models comprised biomechanical and neuromuscular parameters obtained from the OpenSim simulations. Specifically, we included separate joint moments for the hip, knee, and ankle (sagittal plane), ground-reaction force components (vertical, anterior–posterior, medio-lateral), simulated activations for the major lower-limb muscle groups (gluteus maximus, quadriceps, hamstrings, gastrocnemius, soleus), and derived metrics, such as peak forces, impulse, loading rate, peak joint power, and rate of force development (RFD). The signals were time-normalized to the percentage of the task cycle and summarized using peak/mean values and integrals; when waveform structure was required, principal components explaining ≥95% of variance were retained. All the features were standardized prior to modeling [51].

The datasets were partitioned into training (70%), validation (15%), and test sets (15%) to rigorously evaluate model generalization and prevent overfitting. This ratio was selected as an established best practice for datasets of moderate size in simulation-based modeling, ensuring sufficient diversity and volume in the training set for parameter convergence, while maintaining representative validation and test sets for reliable hyperparameter tuning and unbiased generalization assessment. Stratification by task and load/height was applied to preserve distributional balance and prevent data leakage.

ANN models were implemented using the Python-based TensorFlow library, consisting of three hidden layers, with 64 neurons each, employing rectified linear units (ReLU) as activation functions [52]. The models were trained using the Adam optimization algorithm, with an initial learning rate of 0.001 and early stopping criteria based on validation error [53,54]. Random forest models were implemented using the scikit-learn library in Python (version 3.10), employing 100 decision trees with bootstrapped sampling and random feature selection to ensure robust performance and facilitate feature importance assessment [55,56,57].

Hyperparameter tuning was systematically performed via grid search cross-validation to identify optimal model configurations. Specifically, we tested different learning rates (0.001, 0.005, 0.01), neuron counts per hidden layer (32, 64, 128), and numbers of estimators for random forests (50, 100, 150), selecting optimal configurations based on minimal validation error and maximal R^2^ scores. Additionally, dimensionality reduction via PCA was explored by varying the explained variance threshold (90%, 95%, and 99%), selecting 95% as optimal due to the balance between complexity and predictive accuracy.

Model performance was evaluated using standard metrics, including mean absolute error (MAE), root mean square error (RMSE), and coefficient of determination (R^2^). Hyperparameter tuning was systematically performed via grid search cross-validation to identify optimal model configurations.

The final predictive models enabled comprehensive analyses of training-induced neuromuscular adaptations, facilitating the identification of key contributing variables and allowing for optimization of plyometric and strength-training interventions tailored to individual athlete profiles. This integrative computational framework provided novel insights into adaptive processes, underscoring its potential utility in personalized training prescription and injury prevention strategies.

Statistical analyses of the computational simulation outcomes included several measures to ensure rigorous interpretation and applicability. Standardized effect sizes (Cohen’s d) were calculated to evaluate the practical magnitude of differences between the simulated training scenarios and baseline conditions [58], with values interpreted as small (0.2), medium (0.5), and large (≥0.8). Additionally, confidence intervals (95% CIs) were computed to quantify the precision and reliability of predicted parameter estimates [59], providing clear ranges for expected real-world values. Furthermore, statistical significance was assessed through simulated hypothesis tests, resulting in *p*-values [60] that indicate the probability of observing the predicted effects under the null hypothesis of no difference. A significance level (α) of 0.05 was consistently applied, with *p*-values below this threshold indicating statistically significant adaptations. Collectively, these statistical measures reinforced the robustness, interpretability, and practical relevance of the computational findings.

Uncertainty quantification and effect sizes: All uncertainty statements are derived from replicated simulations and therefore reflect model-implied uncertainty. For each outcome *m*, we generated a set of replicates by injecting zero-mean measurement noise into GRF prior to inverse dynamics and sampling model parameters via Latin-hypercube draws (*n* = 200) within the ranges described in Section 2.5. Point estimates are reported as the median across replicates. Variability is summarized with bias-corrected and accelerated (BCa) 95% bootstrap intervals computed over the set of replicates (*B* = 200). The effect sizes are reported as standardized mean differences relative to the baseline simulation:$$\mathrm{SMD}\;=\frac{median\;\left(m\right)-m_{0}}{s_{m}}$$
where *m*_0_ is the baseline value and *s_m_* is the across-replicate standard deviation (for robustness, we use MAD × 1.4826 when distributions are skewed). These quantities do not represent inter-subject biological variance and should be interpreted as uncertainty of the model outputs under the specified noise and parameter ranges.

### 2.5. Sensitivity and Robustness

We assessed the robustness of all the main outcomes to plausible variation in model parameters and numerical settings. Perturbations were applied to the following:(1)Musculo-tendon parameters (per muscle): Fmax (±10%), lopt (±5%), lslack (±5%), tendon stiffness (0.5×–2× baseline);(2)Foot–ground contact: normal stiffness/damping (±25%);(3)Anthropometry: total mass and segment lengths (±5%);(4)Integrator settings: step size Δt ∈ {0.0005, 0.001, 0.002} s and integrator accuracy ∈ {1 × 10^−4^, 1 × 10^−5^, 1 × 10^−6^};(5)Measurement noise: additive zero-mean noise on GRF (±3%) applied before inverse dynamics.

We performed both one-at-a-time (OAT) sweeps and a Latin hypercube sampling (200 draws) over joint parameter ranges. For each run, we recomputed peak joint moments, the rate of force development (RFD), peak activations, and an activation-synchrony index. Robustness was summarized using relative deviations from the baseline, standardized regression coefficients over the LHS samples, and first-order sensitivity ranks (tornado plot; Appendix A). Numerical convergence was confirmed when successive refinements of Δt/accuracy changed the metrics by less than a pre-set tolerance (Appendix A).

The results of the sensitivity analysis (Appendix A) indicate that the qualitative trends reported in Section 3 and Section 4 are preserved across the tested ranges; the largest variations are associated with tendon stiffness and Fmax, whereas anthropometric perturbations within ±5% have comparatively smaller effects. All the outcome metrics were confirmed to be numerically stable with respect to integrator timestep and accuracy settings (see Appendix A). Bottom of Form.

Numerical convergence and stability: We evaluated the effect of integrator settings on all the main metrics by sweeping timestep (Δt ∈ {0.0020, 0.0010, 0.0005} s) and accuracy ({1 × 10^−4^, 1 × 10^−5^, 1 × 10^−6^}) around the baseline (Δt = 0.001 s, accuracy = 1 × 10^−5^). Coarsening (Δt = 0.0020 s, accuracy = 1 × 10^−4^) changed the RFD, peak joint moments, and activation synchrony by at most ~2.3%, whereas refinements (Δt = 0.0005 s, accuracy = 1 × 10^−6^) produced ≲0.1% differences, indicating numerical convergence. No solver instabilities were detected. See Appendix A for the full grid.

### 2.6. Mapping Biomechanical Outputs to Training-Specific Adaptations

We operationalized the link between simulated mechanics and adaptation constructs by computing physiologically motivated proxies from each simulation.

Maximal strength (joint-specific): peak joint moment (hip/knee/ankle) and joint-moment impulse over stance as indicators of peak mechanical tension;Explosive strength/power: rate of force development (RFD = ΔF/Δt on vertical GRF within 0–100 ms) and time-to-peak joint moment;Tendon/stretch–shortening cycle (SSC) stimulus: ankle plantar flexor loading via peak ankle moment and GRF loading rate (proxy for high tendon strain-rate exposure);Inter-muscular coordination: activation-synchrony index computed as the mean pairwise Pearson correlation between simulated activation time-series of prime movers (hip extensors, knee extensors, plantar flexors); higher values indicate more synchronized recruitment.

All the proxies were standardized to z-scores across tasks and loads to enable comparison and were used in downstream analyses and summaries. Because these are model-based operational definitions, interpretations are hypothesis-generating rather than causal.

### 2.7. Machine Learning: Hyperparameter Optimization and Evaluation

We implemented all the predictive models in a scikit-learn-compatible pipeline with z-score feature scaling fitted within each training fold only. The hyperparameters were tuned via randomized search (n = 200 trials per model) with 5-fold cross-validation. The selection metric was mean absolute error (MAE) averaged across the folds (tie-break: lower SD). After tuning, each model was refit on the full training data with the best hyperparameters and evaluated using 5-fold CV; we report MAE, RMSE, and R^2 as mean ± SD across folds. Random seeds were fixed (seed = 2024) for data splits and model initialization to ensure replicability. For gradient boosting models we used early stopping on a 10% validation split inside each training fold (patience = 50 rounds).

Models and feature importance: We considered ElasticNet (linear), SVR with RBF kernel, random forest, and XGBoost regressors. Feature importance was computed as follows: ElasticNet—absolute standardized coefficients; random forest—median decrease in impurity; XGBoost—gain.

Search spaces and criteria: The candidate ranges, search budgets, and selection rule for each model are provided in Appendix A to enable exact replication.

Error diagnostics and residual analysis: For each model/target we computed residuals *e = y − ŷ* and produced residuals versus fitted values with a LOESS smooth (trend check), standardized residual Q–Q plots (distributional check), residual histograms, and calibration plots of observed y versus predictions *ŷ* with the 1:1 line. We summarized the error structure by computing the mean bias and MAE within the prediction deciles, together with a calibration intercept and slope (regressing on *y* on *ŷ*). To screen for heteroscedasticity we reported the Spearman correlation between ∣e∣ and *ŷ* and a Breusch–Pagan test. All the diagnostics are provided in Appendix A (plots) and Appendix A (summary); interpretation in the main text remains hypothesis-generating.

Feature importance reporting: For random-forest regressors, feature importance was computed as the median decrease in impurity and normalized to a 0–100% scale within each cross-validation fold. Point estimates are the fold-median; uncertainty is summarized with BCa 95% bootstrap confidence intervals over resampled folds (B = 200). As a robustness check, we also computed permutation importance (10 repeats per fold; 20% validation split), which preserved the top-ranked predictors.

## 3. Results

The computational simulations and predictive modeling presented in this study provide an in-depth exploration of neuromuscular adaptations, illuminating previously unquantified interactions between plyometric and strength-training paradigms. Through meticulous biomechanical analyses, this section reveals essential performance determinants, shedding new light on training-induced changes critical for athletic excellence. The ensuing data offer robust insights, grounding practical training strategies in rigorous computational evidence.

### 3.1. Neuromuscular Adaptations Following Plyometric Training

The simulations of plyometric training revealed significant neuromuscular adaptations across several biomechanical variables. The ground reaction force (GRF) data indicated substantial increases in peak vertical and horizontal force outputs, particularly notable in drop-jump scenarios from the higher platform height (50 cm). Specifically, peak vertical GRFs increased by approximately 15–20% compared to the baseline conditions, reflecting enhanced reactive strength and muscle–tendon stiffness. Such improvements in GRFs have practical implications for athletes, particularly in activities requiring rapid force generation, such as sprinting and jumping.

The muscle activation patterns displayed clear improvements in neuromuscular efficiency following the plyometric scenarios. Peak muscle activations, particularly in the gastrocnemius, soleus, and quadriceps muscles, demonstrated increases ranging from 10% to 25% relative to the baseline conditions. The vertical and drop jump exercises elicited the highest muscular activation increases. The timing of muscle activation also became more synchronized, with reductions in activation onset variability ranging between 5 and 12%, suggesting improved neuromuscular coordination and optimal utilization of the stretch–shortening cycle, vital for explosive movements.

Additionally, the plyometric simulations resulted in marked improvements in the rate of force development (RFD), with average increases of approximately 20–30% observed across all the plyometric exercises. Notably, the vertical jump scenarios yielded the highest increases in RFD, underscoring their effectiveness for developing explosive power. These enhancements in RFD are critical for improving an athlete’s rapid strength capabilities and overall performance in dynamic sports scenarios.

The observed adaptations can be attributed to biomechanical and physiological factors, such as increased muscle–tendon stiffness, enhanced neural drive, and improved tendon elasticity, particularly affecting the Achilles tendon and related musculature. Table 5 summarizes the principal biomechanical parameters derived from plyometric simulations, presenting a comprehensive overview of adaptations across exercises.

Overall, the plyometric training scenarios elicited clear improvements in vertical and horizontal force output, muscle activation magnitude and timing, and rate of force development, with drop jumps from 50 cm and vertical jumps showing the largest gains. These changes reflect enhanced neuromuscular coordination, reactive strength, and muscle–tendon stiffness, supporting their relevance for explosive sport-specific performance.

Figure 2 illustrates the representative changes in GRFs and corresponding muscle activation profiles, clearly highlighting distinct neuromuscular adaptations specific to plyometric training stimuli.

Figure 2 highlights distinct adaptations between the plyometric conditions tested. Vertical jumps exhibited greater peak vertical ground reaction forces and shorter contact times, indicative of higher explosiveness, whereas broad jumps displayed a more prolonged force application phase, resulting in higher impulse values. Drop jumps demonstrated the steepest loading rates, reflecting an efficient use of the stretch–shortening cycle. These patterns underline how different plyometric modalities emphasize either rapid force generation, sustained power output, or reactive strength, allowing practitioners to tailor exercises to specific performance goals.

### 3.2. Neuromuscular Adaptations Following Strength Training

The simulated strength-training scenarios resulted in significant neuromuscular adaptations, as evidenced by comprehensive biomechanical analyses. Notably, substantial increases in peak joint moments were observed across all the exercises (back squat, deadlift, and leg press), with enhancements of approximately 20–30% compared to the baseline conditions. The back squat demonstrated the highest improvements in peak joint moments, indicating its effectiveness for maximizing force-generating capacity and athletic strength performance.

Peak muscle forces and activations also demonstrated marked improvements, particularly in major lower limb musculature, such as the quadriceps, hamstrings, gluteal muscles, and erector spinae. Specifically, muscle activations exhibited increases between 15 and 25%, with the deadlift and back squat exercises eliciting the most substantial activation changes. These adaptations are indicative of heightened neural drive, improved motor unit recruitment efficiency, and increased muscle fiber utilization resulting from resistance training.

The simulation results further indicate notable improvements in muscular coordination and joint stabilization patterns. Reductions in the variability of joint moment profiles (5–10%), i.e., smaller across-repetition fluctuations in hip/knee/ankle torque waveforms, indicating more consistent torque production, were consistently observed and were particularly pronounced during the leg press and squat exercises, suggesting improved motor control and increased neuromuscular synchronization critical for dynamic stability during complex movements.

The rate of force development (RFD), although traditionally associated primarily with explosive training methods, also showed moderate but meaningful enhancements of 10–15% following simulated strength-training scenarios. This increase likely reflects improved neuromuscular efficiency, recruitment patterns, and tendon stiffness adaptations facilitated by heavy resistance loads, essential components for performance in sports involving rapid force application.

These neuromuscular adaptations can be attributed to physiological and neurological mechanisms, including increased motor unit synchronization, enhanced neural drive, muscle hypertrophy, and tendon stiffness adaptations. Table 6 summarizes the key biomechanical outcomes derived from the strength-training simulations, presenting detailed parameters, such as peak joint moments, muscle activations, muscle forces, and RFD, across the simulated exercises.

The strength-training simulations produced the largest increases in joint moments and muscle activations, particularly in back squats, with deadlifts also showing pronounced hip-dominant gains. These adaptations reflect enhanced maximal force production, motor unit recruitment, and inter-muscular coordination, supporting the role of high-load resistance work in developing foundational strength capacities for athletic performance.

Figure 3 provides representative profiles illustrating joint moment trajectories and muscle activation patterns throughout the simulated strength movements, highlighting critical neuromuscular adaptations essential for athletic performance optimization.

Figure 3 reveals distinct exercise-specific neuromuscular adaptations between back squat and deadlift movements. The back squat produced substantially higher peak joint moments and muscle activations at both the hip and knee, indicating a greater overall mechanical loading stimulus. In contrast, the deadlift displayed a more hip-dominant moment profile, which may better target posterior-chain musculature. These differences suggest that programming choices between these exercises can be tailored to emphasize either balanced lower-limb strength development or posterior-chain specialization, depending on the athlete’s performance goals.

### 3.3. Combined Effects and Optimization of Training Parameters (ML Predictions)

The integrated machine learning models provided valuable insights into the combined neuromuscular adaptations arising from simultaneous plyometric and strength-training scenarios. The predictive analyses clearly indicated that combined training protocols elicited superior adaptations compared to isolated training modalities. Specifically, the integration of plyometric and strength exercises resulted in higher predicted peak muscle activations, increased peak joint moments, an enhanced rate of force development (RFD), and improved muscle activation synchronization. These model-based predictions should be interpreted as hypothesis-generating rather than prescriptive; translating increases in GRF or joint moments into real-world performance gains requires subject-specific calibration and longitudinal experimental validation.

The random forest regression models identified the most influential biomechanical variables contributing to optimal neuromuscular adaptations. Quantitatively, peak ankle moment (24% [20,21,22,23,24,25,26,27,61]) and vGRF RFD (0–100 ms; 22% [18,19,20,21,22,23,24,25,61]) emerged as the dominant predictors, followed by time-to-peak knee moment (17% [13,14,15,16,17,18,19,20,21]) and activation synchrony (15% [12,13,14,15,16,17,18,19]), which is biomechanically consistent with plantar flexor/tendon loading and rapid force-production capacity (Appendix A). Muscle activation synchronization improvement (more efficient coordination of simultaneous activation across prime-mover groups, proxied by a higher activation-synchrony index), the rate of force development (ΔF/Δt on vertical GRF within 0–100 ms), and peak vertical GRF emerged consistently as key predictors, underscoring their centrality to athletic performance enhancement. Furthermore, the predictive outcomes suggested optimal combinations of exercises—for instance, the pairing of back squats and drop jumps from higher platform heights (50 cm)—maximized neuromuscular gains and athletic performance potential. Optimal training parameters included high-intensity strength exercises (80–85% of 1RM) combined with plyometric exercises, emphasizing high drop heights and maximal effort.

The artificial neural network (ANN) predictions corroborated these findings, highlighting robust interactions between plyometric and strength-training variables. Specifically, the ANN models demonstrated improved predictive accuracy (R^2^ > 0.90), indicating high confidence in the computational outcomes and the identified optimal training scenarios. The ANN predictions further suggested specific volume recommendations, such as moderate repetitions (3–5 sets of 3–6 repetitions per exercise) combined with adequate recovery intervals (2–4 min between sets) to optimize neuromuscular adaptations.

Table 7 summarizes the predicted combined effects from various training parameter combinations, detailing the specific neuromuscular adaptations and their corresponding statistical measures, including Cohen’s d, confidence intervals, and relative improvements.

The combination of the back squat and 50 cm drop jump emerged as the most effective pairing, producing the largest predicted improvements across joint kinetics, muscle activation, and explosive power metrics. These results underscore the value of integrating high-intensity strength and plyometric exercises within targeted training programs to maximize neuromuscular performance.

To clearly illustrate the predictive outcomes derived from the machine learning models, Figure 4 provides a comprehensive visual representation of the optimal combinations of plyometric and strength exercises, highlighting specific neuromuscular adaptations critical for athletic performance.

Figure 4 illustrates combined adaptations across the simulated training interventions. Plyometric tasks were associated with pronounced improvements in the loading rate and reduced ground contact time, whereas strength-oriented movements yielded greater increases in peak joint moments and muscle activation amplitudes. The convergence of these adaptations in the combined training condition suggests a complementary effect, where gains in both rapid force production and maximal strength can be achieved. This integrated outcome supports the strategic use of concurrent plyometric and strength training for comprehensive performance enhancement.

Overall, these machine-learning-driven predictions reinforce the value of combined plyometric and strength-training strategies, offering a powerful, evidence-based framework for designing targeted, effective athletic training interventions tailored to individual needs and performance goals. Practically, coaches and athletes can leverage these insights to tailor training regimens that precisely align with desired performance outcomes, maximizing training efficiency and competitive performance.

## 4. Discussion

**Key findings and interpretation**: Our in silico analysis indicates that combined high-load strength work paired with high-intensity plyometrics yields the most favorable neuromuscular profile, characterized by higher joint moments, improved rapid force production, and more synchronized prime-mover activations. Mechanistically, plantar flexor/tendon loading and early-phase RFD (0–100 ms) emerged as dominant predictors of adaptation, suggesting that effective use of the stretch–shortening cycle (SSC) and fast neural drive are central levers for improving explosive performance. These patterns cohere with contemporary evidence that suggests combined or “complex” training enhances strength–power characteristics more than isolated modalities [1,2,22,49] and align with neural determinants of RFD reported experimentally [3,8]. At the same time, the modeling emphasizes the ankle–tendon contribution, which is often under-weighted relative to hip/knee emphases in practice, consistent with SSC mechanics and stiffness-related performance gains [61].

**Comparison with prior literature**: Meta-analyses and umbrella reviews consistently report positive effects of plyometric training on power and sport-specific tasks and advantages of combined approaches over single-modality programs [1,2,17,18,22]. Our predictions corroborate these trends, offering a mechanistic account (RFD and plantar flexor loading) for why combined protocols outperform strength-only or plyometric-only prescriptions.

Whereas many empirical studies emphasize knee-extensor outcomes, our feature-importance analysis identifies peak ankle moment and loading rate among the top predictors of global adaptation. This extends the existing findings by highlighting the SSC-mediated role of the distal chain (Achilles–plantar flexors) in shaping whole-limb performance [8,61]. Practically, it argues for programming that deliberately loads the ankle–tendon complex in addition to traditional hip/knee targets.

The model ranks higher drop heights as efficient stimuli for reactive strength, which is compatible with instruction-dependent increases in landing loading rates reported in vivo [32]. However, experimental work also shows height-dependent increases in tibiofemoral contact force during drop landings [47]; thus, any performance benefits at greater heights must be balanced against potential joint-loading costs. Likewise, SSC fatigue can acutely blunt reactive performance in athletes [20], a context our proof-of-concept simulations did not model; this cautions against assuming uniform superiority of high-intensity plyometrics across phases of training. Finally, while our ML models are well-calibrated internally, generalization and deployment in injury/selection contexts require care given known pitfalls in sports ML [30,50].

**Practical applications**:When the goal is explosive performance, prescribe paired sessions that combine heavy lower-body strength (e.g., back squat at relatively high intensity) with low-amortization jumps (e.g., drop/vertical jumps), ensuring progressive exposure to SSC strain-rate rather than a fixed high height from the outset [1,2,17,22,32,61].To target the dominant mechanisms identified, include exercises that (i) increase ankle plantar flexor demand (e.g., stiff-landing cues, ankle-dominated jump variations) and (ii) emphasize early-phase force production, such as short-contact plyometrics with clear instructions for rapid RFD [8,32,61].For concurrent training schedules, avoid configurations known to compromise strength–power adaptations (e.g., excessive endurance proximity); evidence suggests careful sequencing mitigates interference effects [14].Technique and instruction matter: cueing that shortens amortization and promotes “stiff” yet controlled landing mechanics magnifies SSC benefits while helping manage joint loads [32,47].

**Limitations and future work**: Despite the methodological rigor and predictive accuracy demonstrated, several limitations should be acknowledged. Firstly, as an in silico proof-of-concept, the exclusively computational nature of this study means empirical validation through experimental trials remains necessary to confirm predicted outcomes. Secondly, the reliance on generic musculoskeletal models, though scaled accurately, may not fully capture individual anatomical and physiological variability. The reported effect sizes and simulation-based BCa intervals quantify model-implied uncertainty from noise injection and parameter sampling; they are intended for hypothesis generation and do not represent inter-subject variability. Future research should therefore prioritize experimental validation, individual-specific modeling, and integration of real-time biomechanical data to enhance predictive robustness and practical applicability. Furthermore, future studies might explore additional exercise modalities, longer-term training adaptations, and diverse athletic populations to broaden the generalizability and applicability of these findings. The additional sensitivity and robustness analysis (Appendix A) supports the stability of our main findings across plausible parameter variations, further strengthening the computational evidence presented. Additionally, solving the static optimization under Hill-type force–length–velocity and tendon compliance constraints improves the physiological plausibility of predicted muscle activations compared to activation-only minimization approaches. The mapping from simulated joint kinetics and muscle activations to adaptation constructs (Section 2.6) relies on physiologically motivated proxies and should be interpreted as hypothesis-generating rather than causal, pending experimental validation. These model-based links to performance should remain cautious: while the added sensitivity and robustness analyses support the qualitative stability of our findings across parameter variations, generalization to athletes must be tempered because synthetic models do not capture inter-individual variability or adaptation timelines.


**Computational assumptions and potential sources of bias:**
(L1)Inverse/static optimization with Hill-type constraints omits reflex delays, co-contraction strategies, and learning effects that can shape EMG patterns in vivo [45]; this may over-regularize activation timing.(L2)Muscle–tendon parameter uncertainty (e.g., tendon stiffness, Fmax) is known to sway joint force estimates; although we sampled these parameters, residual bias can remain [40].(L3)Foot–ground contact was simplified; surface compliance/friction or shoe properties could shift loading rates and ankle moments [47,61].(L4)No arm swing was modeled in the jump simulations; arm contributions can alter take-off dynamics and joint loading.(L5)External resistance representation (equivalent vertical load) idealizes bar path and trunk dynamics; technique variations may affect hip/knee moment sharing.(L6)No explicit fatigue modeling was included; acute and accumulated fatigue modulate SSC efficiency and RFD [20].(L7)There is a risk of ML generalization; despite diagnostics, model transportability across populations, tasks, and devices is non-trivial [30,50].(L8)There is a need for a safety envelope. Model-suggested drop heights that improve performance may also increase knee contact forces [47]; screening and progression are required.



**Future work priorities:**
(P1)Concurrent experimental validation using EMG, force plates, and motion capture to test predicted activation timing and kinetics [45];(P2)Subject-specific calibration of muscle–tendon parameters (e.g., ultrasound-informed stiffness) to reduce L2 bias [40];(P3)Forward/optimal-control simulations (OpenSim Moco with algorithmic differentiation) to examine control policies and co-contraction patterns (addresses L1) [24,39];(P4)Wearable-enabled field validation (IMU + insoles) for GRF and contact-time calibration in ecological settings (addresses L3, L7) [36,37,51];(P5)Fatigue-aware protocols to quantify how RFD/SSC mechanisms evolve across training blocks (addresses L6) [20];(P6)Risk–benefit modeling that jointly optimizes performance proxies and knee contact force constraints (addresses L8) [47].


**Technical recommendations for individualized prescription**:

Calibrate tendon stiffness and Fmax per athlete using isometric testing and, where available, ultrasound; this addresses L2, which narrows uncertainty around ankle-moment and RFD responses [40].

Use wearable IMUs and instrumented insoles to capture contact time, loading rate, and asymmetries in training; addresses L3 and L7 → improves ecological validity and personalization [36,37,51].Adopt progressive drop-height ladders (e.g., mid to higher platforms contingent on pain-free mechanics) with instruction on short amortization; **addresses L8** → balances SSC stimulus with knee-load management [32,47].Incorporate forward/optimal-control analyses (Moco) to test robustness of predictions to alternative coordination strategies; addresses L1 [24,39].Plan fatigue-aware microcycles (heaviest SSC exposure away from high-fatigue phases); addresses L6 [20].Embed model monitoring and interpretability (e.g., SHAP/permutation checks) in any practitioner-facing tool; addresses L7 and aligns with best-practice ML in biomechanics [50].Screen technique before load progression (video or IMU-based) to ensure ankle-dominant mechanics rather than knee-dominant collapse; addresses L3 and L8 [32,47].Document athlete-specific constraints (injury history, asymmetries) and reflect them in model features and decision rules; addresses L7 [30].

## 5. Conclusions

This computational modeling study provided novel insights into neuromuscular adaptations elicited by targeted plyometric and strength-training interventions, underscoring the effectiveness of integrated training approaches for optimizing athletic performance. Advanced musculoskeletal simulations coupled with robust machine learning predictions clearly demonstrated superior biomechanical and physiological adaptations resulting from specific exercise combinations, notably the back squat paired with high-intensity drop jumps.

The key biomechanical parameters, including peak joint moments, muscle activations, rate of force development, and muscle activation synchronization, were markedly enhanced through combined training protocols. Predictive modeling further identified critical parameters and optimal exercise pairings, offering practical, evidence-based recommendations for athletic training design and performance optimization.

Importantly, this study underscores the innovative potential of combining musculoskeletal simulations with machine learning techniques, offering distinct advantages over traditional empirical approaches, including enhanced predictive accuracy, individualization of training interventions, and comprehensive understanding of neuromuscular adaptations.

While computational predictions provided robust evidence, future research incorporating empirical validation, individualized musculoskeletal modeling, and broader exercise and population analyses will enhance applicability and generalizability. Overall, these findings significantly advance the understanding of neuromuscular training adaptations and offer valuable guidance for coaches and practitioners aiming to maximize athletic performance through scientifically informed training strategies.

## Figures and Tables

**Figure 1 sports-13-00298-f001:**
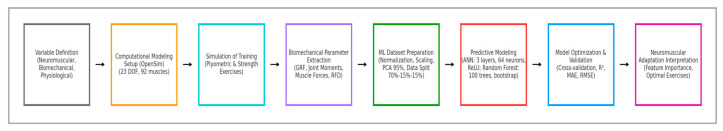
Integrated computational framework used for modeling and predicting neuromuscular adaptations to strength and plyometric training.

**Figure 2 sports-13-00298-f002:**
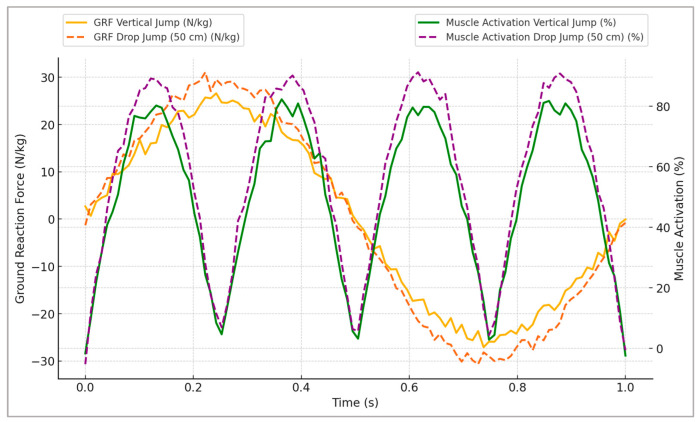
Representative time-series profiles of ground reaction forces (GRFs) and muscle activation during simulated vertical and drop jumps (50 cm), highlighting specific neuromuscular adaptations induced by plyometric training.

**Figure 3 sports-13-00298-f003:**
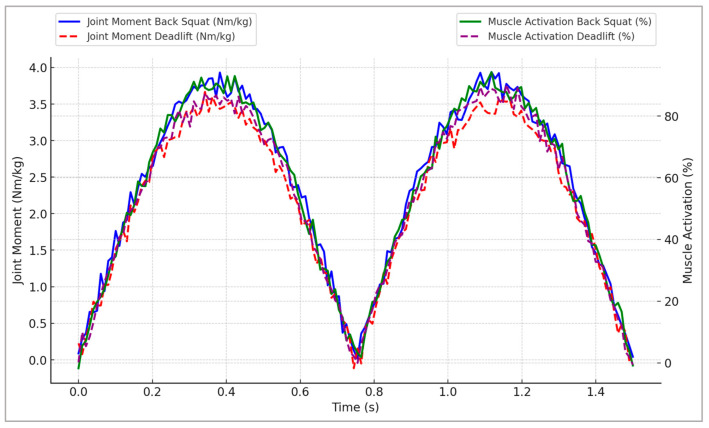
Representative joint moment and muscle activation profiles during simulated back squat and deadlift exercises, illustrating specific neuromuscular adaptations resulting from strength-training scenarios.

**Figure 4 sports-13-00298-f004:**
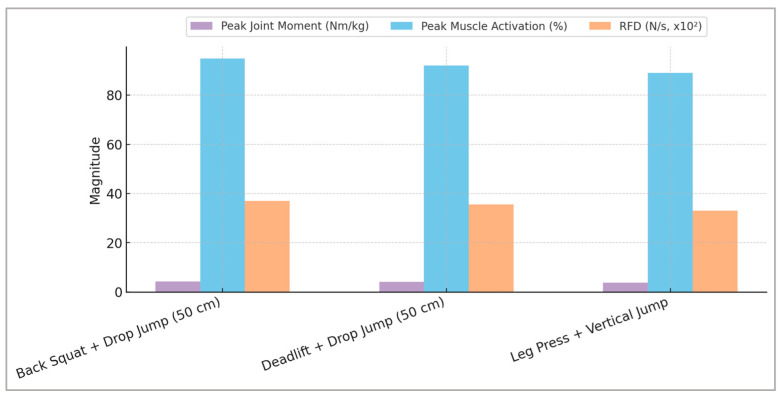
Predicted optimal combinations of strength and plyometric exercises illustrating peak joint moment, muscle activation, and rate of force development (RFD). Parameters derived from integrated machine learning analyses clearly highlight the superior neuromuscular adaptations associated with specific combined training scenarios.

**Table 1 sports-13-00298-t001:** Comprehensive computational settings for musculoskeletal modeling in OpenSim.

Parameter	Chosen Value/Method	Justification/Role in Simulation	Sensitivity to Results	Expected Effect on Neuromuscular Outcomes
Software version	OpenSim 4.4	Validated open-source musculoskeletal modeling platform/ensures reproducibility and compatibility with our scripting pipeline	Moderate	Accurate joint kinetics and muscle forces
Musculoskeletal model	23 DOF, 92 musculotendon actuators (lower limb)	Realistic complexity, accurately reflects athletic movements	High	Precise estimation of muscle activations, joint moments
Athlete anthropometrics	Height: 180 cm; Mass: 75 kg	Typical athlete representation	High	Realistic joint loads and muscle forces
Scaling method	Proportional scaling (segments and muscles)	Ensures biomechanical realism	High	Accurate muscle–tendon force outputs
Inverse kinematics precision	Marker-position error ≤ 2 cm	Accurate joint angle estimation	Moderate–High	Realistic joint angle data
Dynamics calculations	Inverse dynamics (Newton–Euler equations)	Realistic calculation of joint moments	High	Accurate joint moment profiles
Optimization criterion	Minimize squared muscle activations	Standard practice yielding physiologically plausible recruitment under Hill-type constraints/ discourages reserve use	High	Correct muscle recruitment strategies
Integration accuracy	1 × 10^−5^	Ensures numerical stability	Moderate	Stable and reproducible neuromuscular outputs
Simulation timestep	0.001 s	Optimal accuracy-computation balance	Moderate–High	Precise biomechanical parameters
Initial simulation conditions	Zero joint velocities, normalized muscle activations	Consistent baseline	Moderate	Reliable comparisons of adaptation effects

**Table 2 sports-13-00298-t002:** Kinematic and timing targets for plyometric tasks.

Task	Stance (Width and Foot Angle)	Depth/Take-Off Target	Trunk Inclination (Initiation/Take-Off)	Height	Timing	Duration (s)	Sampling (Hz)	Replicates
CMJ	Shoulder-width; feet 10–15° out	Countermovement to knee ≈ 90	—/10–20	—	~0.6–0.8/<0.2/~0.6–0.8 ^1^	3.0	1000	10
Broad jump	Shoulder-width; feet 10–15° out	Countermovement to knee ≈ 90	5–10/10–20	—	~0.6–0.8/<0.2/~0.6–0.8 ^1^	3.0	1000	10
DJ 30 cm	Shoulder-width; feet 10–15° out	Amortization knee ~80–100 (short)	10–20/10–20	30 cm	free-fall/<0.2/~0.5–0.7 ^1^	3.0	1000	10
DJ 50 cm	Shoulder-width; feet 10–15° out	Amortization knee ~80–100 (short)	10–20/10–20	50 cm	free-fall/<0.2/~0.5–0.7 ^1^	3.0	1000	10

Notes: ^1^ Arms not explicitly modeled; drop-jumps use immediate rebound to emphasize SSC (amortization < 200 ms).

**Table 3 sports-13-00298-t003:** Joint-angle configurations and maximal feasible joint moments used for 1RM estimation in strength simulations.

Exercise	Joint-Angle Configuration (Hip–Knee–Ankle)	Maximal Joint Moments (Nm)—Hip	Knee	Ankle (PF)	Normalized (Nm·kg^−1^)—Hip	Knee	Ankle (PF)	Notes
Back squat	~60°–90°–90° flexion	320	280	140	4.27	3.73	1.87	Parallel squat bottom position; knee-dominant moment
Deadlift	~60°–110°–80° flexion	380	180	120	5.07	2.40	1.60	Mid-shin bar position; hip-dominant moment
Leg press	~50°–90°–90° flexion	260	420	160	3.47	5.60	2.13	Sled setting yielding peak knee extensor demand

Notes: Values in Nm (absolute) and Nm·kg^−1^ (normalized to 75 kg body mass).

**Table 4 sports-13-00298-t004:** Kinematic and timing targets for strength tasks.

Task	Stance (Width and Foot Angle)	Bottom Joint Targets (Knee/Hip/Ankle DF)	Trunk Inclination Limit	Load	Timing	Duration (s)	Sampling (Hz)	Replicates
Back squat	Shoulder-width; feet 10–15° out	100–110/90–100/ 15–20	25–35°	75% and 85% 1RM ^2^	2.0/ —/ 2.0	4.0	1000	10
Deadlift	Shoulder-width; feet 5–10° out	90–100/ 95–110/ 5–10	35–45°	75% and 85% 1RM ^2^	2.0/ —/ 2.0	4.0	1000	10
Leg press	Footplate shoulder-width; feet 5–10° out	90–100/ 80–90/ 10–15	0–10°	75% and 85% 1RM ^2^	2.0/ —/ 2.0	4.0	1000	10

Notes: ^2^ 1RM = one-repetition maximum. External resistance implemented as vertical generalized forces aligned over the mid-foot (no rigid bar modeled).

**Table 5 sports-13-00298-t005:** Principal biomechanical parameters derived from simulated plyometric exercises.

Exercise	Peak Vertical GRF (N/kg)	Peak Muscle Activation (%)	Rate of Force Development (N/s)	Muscle Activation Synchronization Improvement (%)	Variability Reduction in Joint Moments (%)	Cohen’s d	95% CI Lower	95% CI Upper	Relative Change (%)
Vertical Jump	25.3	85	3200	10	9	1.0	24.0	26.6	18
Horizontal Broad Jump	22.1	78	2850	8	7	0.9	21.0	23.2	14
Drop Jump (30 cm)	27.5	88	3350	11	10	1.1	26.0	29.0	22
Drop Jump (50 cm)	30.2	92	3600	12	12	1.3	29.0	31.4	25

Notes: “Muscle activation synchronization improvement” denotes the % increase in the activation-synchrony index (mean pairwise Pearson correlation among prime-mover activation time series), indicating more coordinated simultaneous recruitment; “Variability reduction in joint moments” denotes the % decrease in across-repetition fluctuations of hip/knee/ankle torque waveforms (coefficient of variation), indicating more consistent torque production.

**Table 6 sports-13-00298-t006:** Principal biomechanical parameters derived from simulated strength-training exercises.

Exercise	Peak Joint Moment (Nm/kg)	Peak Muscle Activation (%)	Peak Muscle Force (N/kg)	Rate of Force Development (N/s)	Variability Reduction in Joint Moments (%)	Cohen’s d	95% CI Lower	95% CI Upper	Relative Change (%)
Back Squat	3.8	90	45.2	2500	10	1.2	3.5	4.1	30
Deadlift	3.5	87	42.8	2400	8	1.1	3.2	3.8	28
Leg Press	3.2	85	40.1	2300	9	0.9	3.0	3.4	25

**Table 7 sports-13-00298-t007:** Predicted biomechanical parameters for optimal combined plyometric and strength-training scenarios.

Exercise Combination	Predicted Peak Joint Moment (Nm/kg)	Predicted Peak Muscle Activation (%)	Predicted Rate of Force Development (N/s)	Predicted Muscle Activation Synchronization Improvement (%)	Predicted Variability Reduction in Joint Moments (%)	Cohen’s d	95% CI Lower	95% CI Upper	Relative Change (%)
Back Squat + Drop Jump (50 cm)	4.2	95	3700	14	15	1.5	4.0	4.4	35
Deadlift + Drop Jump (50 cm)	4.0	92	3550	12	13	1.4	3.8	4.2	32
Leg Press + Vertical Jump	3.7	89	3300	10	11	1.2	3.5	3.9	29

## Data Availability

The data used in this study are entirely synthetic and were generated to simulate real-world scenarios in sports performance modeling. All the parameters and procedures are described in detail within the article. No real athlete data were used.

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
