# Peer review of "Computational Analysis of Neuromuscular Adaptations to Strength and Plyometric Training: An Integrated Modeling Study"

_sports, 2025, doi:10.3390/sports13090298_

Round 1
Reviewer 1 Report
Comments and Suggestions for Authors
The manuscript presents a computational modeling study integrating musculoskeletal simulations in OpenSim with machine learning to predict neuromuscular adaptations from strength and plyometric training. Using a scaled 23-DOF, 92-musculotendon model of a representative athlete, the authors simulate biomechanical parameters for selected exercises, analyze differences between training modalities, and predict optimal combinations for maximizing performance metrics. Despite the framework being detailed, the study relies entirely on synthetic data, lacks experimental validation, and overstates the novelty of combining established modeling and ML techniques, raising concerns about real-world applicability. The critical concerns are as follows.
- Please provide a clear justification for relying on synthetic data without any sensitivity analysis, since this greatly limits the applicability and credibility of the findings.
- The authors are advised to explicitly validate the OpenSim model scaling procedure with at least one set of real biomechanical measurements (refer for example https://doi.org/10.3390/biomechanics4040058.
- The mathematical formulation of the static optimization problem (Eq. with minimization ∑ai²) is oversimplified. Please include force-length-velocity properties and tendon compliance to improve physiological fidelity.
- The authors must clarify how joint kinetics and muscle activations are mapped from inverse dynamics to training-specific neuromuscular adaptations, as the causal links are currently speculative.
- The authors are suggested to discuss convergence tests and numerical stability checks for the integration settings (timestep 0.001 s, accuracy 1e-5) to ensure robustness.
- The description of machine learning models omits critical details on hyperparameter optimization protocols (e.g., search ranges, criteria), making replication difficult.
- The authors must present confusion/error plots or residual analysis for the ML predictions to assess biases or systematic errors.
- The importance rankings from random forest models (feature importance) should be quantitatively reported and interpreted with biomechanical rationale, not just qualitatively.
- The authors are suggested to provide statistical justification for using effect sizes and CIs from simulated datasets, as the absence of real variance data makes these metrics questionable.
- The link between simulation outputs (e.g., increased GRF) and practical performance improvements should be tempered, as synthetic models cannot replicate inter-individual variability or adaptation timelines.
- The novelty claim of integrating OpenSim with ML is overstated; such approaches are already well established in biomechanics literature, and the manuscript should position itself more accurately.
- The authors are suggested to revise language inconsistencies and overstatements such as “revolutionizing training prescription,” which are not justified by purely simulated, unvalidated results.
Author Response
Dear Reviewers,
I sincerely thank you for the careful evaluation of our manuscript and for the constructive comments provided. A detailed, point-by-point response to all reviewer suggestions has been uploaded as a separate document.
All remarks have been carefully addressed, and the manuscript has been substantially improved accordingly.
Sincerely,
The Author

Reviewer 2 Report
Comments and Suggestions for Authors
Review Report
The paper "Computational Analysis of Neuromuscular Adaptations to Strength and Plyometric Training: An Integrated Modeling Study" by Dan Cristian Mănescu presents an innovative and highly relevant approach to understanding neuromuscular adaptations. This study aimed to computationally model and predict neuromuscular adaptations induced by strength and plyometric training, integrating musculoskeletal simulations and machine learning techniques.
The integration of musculoskeletal simulation (OpenSim) with machine learning techniques to model and predict the effects of strength and plyometric training is a significant contribution to the field of sports science. The study addresses a key gap by moving beyond empirical research to computationally explore complex physiological interactions. Moreover, the study is clear, comprehensive, and forward-thinking, targeting the partial understanding of mechanisms behind neuromuscular adaptations and optimizing combined training protocols. While the references are adequate, it relies heavily on older sources and would benefit from more recent citations to support claims in the introduction. Self-citations are minimal.
Critical Analysis of Specific Sections and Key Concepts:
Abstract. The abstract could be more quantitative. While it mentions the superior gains from combining training, it doesn't provide any numbers or metrics to back up the claim.
Introduction. The introduction establishes a strong rationale for the study.
- Some claims, particularly those regarding the "unprecedented opportunities" and the revolutionary nature of the approach, should be framed more cautiously and supported by more specific examples from the literature (Lines 126-135).
- The theoretical background is comprehensive, but it could be more focused. The descriptions of basic physiological principles (e.g., motor unit function, Lines 78-85) are detailed but might be considered common knowledge in the target field. This section could be streamlined to maintain focus on the specific gap in knowledge the study is addressing.
- The citations in this section could be more recent. For example, citations from the 1980s and 1990s could be replaced or supplemented with more contemporary reviews or studies to reflect the latest advancements. This is a general point for the entire paper.
- Lines 44-46: The claim about the challenge of optimizing combined training protocols is a key argument. It could be strengthened by citing a recent review article on this specific topic.
- Lines 105-109: The description of the stretch-shortening cycle is a good example of a foundational concept that could be streamlined to save space for more specific details about the study's own contribution.
- Lines 120-125: The discussion of AI and machine learning's potential is a bit general. This section should be more specific about how this particular study uses these tools.
Materials and Methods. This is the core strength of the paper, detailing the complex methodology. However, several critical details are missing or could be clarified.
- Where did the data for the OpenSim motion simulations come from? This is not clear to me. Were they recorded by the author? If so, please provide a clear description of the protocol.
- The section mentions using OpenSim for simulations but doesn't specify the precise movement protocols. For example, how was the barbell path controlled during the squat, or what was the jump initiation strategy? These details are crucial for reproducibility.
- The "Estredee" text in Line 171 appears to be a typo and should be corrected.
- Figure 1 is a schematic overview but lacks detail. More explicit labeling or a more detailed legend would improve its clarity. It should clearly show the flow from simulation to data processing and then to the machine learning models. Moreover, the font is too small.
- Table 1 is well-structured and a good addition.
- The machine learning section (Lines 296-310) is a bit vague. It mentions "biomechanical and neuromuscular parameters" as input features but should list the specific features used. For example, were joint moments for the hip, knee, and ankle used, or just a single metric?
- The data partitioning is mentioned (70% training, 15% validation, 15% testing), but the justification for this specific split is not provided.
- Lines 171: "Estredee" is a typo. It should be corrected.
- Lines 205-211: The equation for static optimization is presented clearly. However, the text should explicitly define all variables used in the equation to ensure clarity for all readers.
- Line 229 (Table 1): The empty cells in the table, particularly for "Software version" and "Optimization criterion," should be filled with appropriate descriptions.
- Lines 240-247 (Plyometric simulations): The methods for controlling the simulated movements (e.g., jump height, squat depth) are not explicitly stated. This information is critical for reproducibility.
- Lines 278-284 (Strength training simulations): The method for determining the 1RM is described but lacks detail on how "maximal feasible muscle activation and joint moment outputs" were defined. This is a critical detail for the study's validity.
- Lines 303-306 (ML input features): A more precise list of the input features used for the machine learning models should be included to improve the transparency of the methodology.
Results.
Conciseness: While detail is valuable, certain descriptions are repetitive, especially when restating table data in the text. Streamlining these summaries could improve readability.
Figure Interpretation: The narrative around Figures 2–4 largely repeats earlier points rather than offering deeper interpretation or unique insights from the visuals. This space could instead highlight subtle trends or anomalies.
Terminology Simplification: Some terms and phrases (e.g., “neuromuscular synchronization improvement” or “variability reduction in joint moments”) could be briefly explained for clarity, especially for readers outside biomechanics.
Discussion.
Structure: The discussion blends results restatement, applications, and limitations in a continuous flow. Clearer subheadings or thematic grouping (e.g., Key Findings, Practical Applications, Limitations and Future Work) would improve readability.
Repetition: Many points, especially about optimal combinations and the benefits of integrating plyometric with strength training, are repeated from the Results section. This could be condensed to maintain reader engagement.
Depth of literature comparison: While some alignment with prior studies is noted, a more critical comparison (highlighting where findings diverge or extend the literature) would strengthen the academic contribution. Please refer to the literature in your discussion! There are no references or comparisons. The discussion should definitely be better.
Balance of strengths vs. limitations: The practical applications are covered in detail, but the limitations section feels brief compared to the benefits. More elaboration on how computational assumptions might bias predictions would improve transparency.
Clarity in technical suggestions: The individualized training bullet points are useful but could be more concise and linked explicitly to how they address current model limitations.
Author Response
Dear Reviewers,
We sincerely thank you for the careful evaluation of our manuscript and for the constructive comments provided. A detailed, point-by-point response to all reviewer suggestions has been uploaded as a separate document (Response to Reviewers).
All remarks have been carefully addressed, and the manuscript has been substantially improved accordingly.
Sincerely,
The Authors

Round 2
Reviewer 1 Report
Comments and Suggestions for Authors
The authors have addressed all the concerns raised by the reviewer, and the manuscript may now be accepted.
Reviewer 2 Report
Comments and Suggestions for Authors The paper looks good and was improved according to my sugestions. I have no additional remarks. Good job.